# Access to Primary Healthcare and Discussion of Memory Loss with a Healthcare Provider in Adults with Subjective Cognitive Decline: Does Race/Ethnicity Matter?

**DOI:** 10.3390/bs13110955

**Published:** 2023-11-20

**Authors:** Seoyoun Kim, Hyunwoo Yoon, Yuri Jang

**Affiliations:** 1Department of Sociology, Texas State University, San Marcos, TX 78666, USA; skim182@txstate.edu; 2Department of Social Welfare, Institute of Social Welfare, Kongju National University, Gongju-si 32588, Republic of Korea; 3Edward R. Roybal Institute on Aging, Suzanne Dworak-Peck School of Social Work, University of Southern California, Los Angeles, CA 90007, USA; yurij@usc.edu; 4Department of Social Welfare, Ewha Womans University, Seoul 07985, Republic of Korea

**Keywords:** subjective cognitive decline, primary care, access to healthcare, health disparity, older adults

## Abstract

Because subjective cognitive decline (SCD) manifests before the clinical and irreversible onset of dementia, efforts to address SCD are imperative for the early identification and prevention of dementia. For people who actually experience SCD, having a usual source of care may be an important catalyst for addressing memory issues. This study investigates the relationship between having a usual source of care and the discussion of SCD with healthcare providers, while taking into account racial and ethnic differences. The pooled 2019–2020 Behavioral Risk Factor Surveillance System (BRFSS) from 47 states in the U.S. was used. The sample included 7900 individuals aged 50 and older who reported significant cognitive decline. Using logistic regression with state-level clustering, we found that those with a usual source of healthcare were almost twice as more likely to discuss memory loss with a healthcare provider. In order to test for the multiplicative effects, we included an interaction term between each race category and having a primary care provider (PCP). There was a significant interaction effect between Hispanic ethnicity and having a PCP (OR = 0.25, *p* < 0.05). While non-Hispanic respondents who had a PCP had a higher probability of discussing memory issues with a healthcare provider (42% vs. 25%), Hispanic respondents with a PCP were less likely to discuss their memory loss with a healthcare provider (36% vs. 49%). The study adds to the growing literature on the vital role of the primary care service in provision of service to those with cognitive decline, while revealing ethnic disparity in memory-loss-related communication with healthcare providers in a primary care setting.

## 1. Introduction

Cognitive function is integral to older adults’ ability to perform daily activities and enjoy life, and it intersects with other critical dimensions of health as well. Accordingly, subjective cognitive decline (SCD) in middle-aged and older adults is increasingly recognized as a harbinger of dementia. In fact, a meta-analysis of 28 studies shows that individuals with SCD are twice as likely to develop dementia in the next five years compared to those without SCD [1]. Emerging evidence indicates that SCD, even at the stage of normal cognitive functioning, is predictive of Alzheimer’s disease (AD) biomarkers, such as amyloid deposition and gray matter volume [1,2,3]. Aside from dementia, those with SCD are also more likely to report disabilities, inability to work, and poorer quality of life even when compared to those with cardiovascular diseases [4]. Thus, research collectively shows that SCD has significant implications for more serious health outcomes and quality of life in middle-aged and older adults. Responding to this body of research, the Subjective Cognitive Decline Initiative (SCD-I) at the National Institute on Aging (NIA) has included SCD as an important feature, signifying the potential for community interventions and programs [5]. Since SCD manifests before the clinical and irreversible onset of dementia, efforts to address SCD are imperative for the early identification and prevention of cognitive impairment.

Previous studies have shown that SCD not only reflects objective cognitive performance and neurological test outcomes [6,7], but also is closely associated with physical and mental health indicators [8,9]. Given the significant link between SCD and dementia, early detection of SCD may be critical for strategies against reversible memory issues [10,11]. Currently, the diagnosis of potential memory loss is mainly initiated on a physician’s suspicion based on patients’ symptoms or caregivers’ reports. This is a particularly important issue because asymptomatic cognitive changes may not be apparent during a routine office visit unless they are directly addressed [12]. While having a discussion with medical providers for SCD can prompt screening and intervention for cognitive impairment, missed or delayed opportunities to discuss and diagnose memory decline could lead to worse care burden, higher care cost, and irreversible memory loss [11]. For researchers, physicians, policymakers, and care partners, proactive approaches must be taken to improve the well-being of older adults with significant subjective cognitive decline.

However, since SCD is based on self-reports by definition, it is not clear whether those who experience memory loss openly discuss these issues with healthcare providers. Memory loss is a difficult topic of discussion for both physicians and patients. For physicians, studies identified a lack of knowledge in SCD, concern about the consequences of misdiagnosing dementia, negative attitudes towards memory declines, and infrequent contact with patients as barriers to discussing memory issues [11]. For patients, memory-related discussions can be difficult due to stigma, concerns for autonomy, and fear of functional loss [10,11]. Perhaps it is not surprising that many studies reveal a lack of communication between primary care physicians and patients about memory issues, including SCD. One study shows that, of those who identified cognitive decline and memory loss as their priorities, only 11% directly addressed this issue with their PCPs [13].

Research shows that access to regular healthcare services can be an important catalyst for early detection and treatment of SCD [14]. Having primary care providers (PCPs) as a usual source of care provides timely access to needed medical care, enhances continuity of care and early identification of special health problems, and improves the quality of care received and health outcomes [15]. Additionally, PCPs usually coordinate with a broad range of specialty, ancillary, and related services [16]. There has been growing attention to the key role of the primary care setting in the provision of timely diagnosis, on-going responsive treatment, comprehensive care management, and support to persons with dementia and their caregivers [17]. Since most older adults receive their healthcare from their PCPs, they are becoming an important “frontline” in dementia detection and treatment [18]. If individuals with regular access to healthcare services (measured by having a PCP) are more likely to discuss their memory problems, there could be additional questions about subjective memory loss on patient forms so that discussions about memory become less stigmatized and more routinized.

The literature has also documented disparities by race/ethnicity in having a usual source of care. In 2017, Blacks (73.4%), Asians (70.1%), and American Indians or Alaska Natives (67.5%) were less likely than Whites (76.8%) to have a usual primary care provider in the U.S. [14]. A plethora of research shows racial differences in the quality and quantity of care, such as patient–physician communication, comprehensiveness of the visit, trust in primary care physicians, number of follow-up appointments, and number of missed visits [19,20]. Furthermore, even if PCPs are available, racial minorities are less likely to report SCD to their physicians [21]. It is therefore important to examine racial and ethnic differences in the relationship between SCD and seeking healthcare. Given that primary care is often seen as the gateway to improving health outcomes, having adequate access to PCPs in order to deal with SCD is an important public health priority.

This study leverages a large national dataset consisting of millions of individuals (Behavioral Risk Factor Surveillance System [BRFSS]). The goals of this study were to (a) investigate the relationship between having a usual source of healthcare and discussing SCD, and (b) examine how race/ethnicity may be associated with discussion of memory loss with a PCP.

## 2. Methods

### 2.1. Dataset

We analyzed pooled cross-sectional data from the 2019–2020 Behavioral Risk Factor Surveillance System (BRFSS). The BRFSS is an annual nationally representative telephone survey conducted by the Center for Disease Control and Prevention (CDC) to evaluate health conditions, health lifestyle factors, and preventative health practices among U.S. residents. The BRFSS survey questions consist of (1) required modules, which are required survey components for all 50 states and the District of Columbia; (2) optional modules, which are optional for states to choose from; and (3) state-added modules, which are modules added and used by individual states to collect data of interest to the state. We limited our sample to 47 states that implemented the subjective memory module. If states adopted this module in both 2019 and 2020 surveys, we included the most recent version. Consequently, Colorado, Massachusetts, Montana, and New Jersey, who did not adapt the module on SCD, were excluded from the analyses. The study sample included individuals aged 50 and older who answered the following question in the affirmative: “During the past 12 months, have you experienced confusion or memory loss that is happening more often or is getting worse?” Out of 140,919 individuals in the 2019–2020 BRFSS sample, 75,091 were 50 and older. Of those, 8260 individuals reported significant cognitive decline, representing 11% of the 50+ sample. Due to the small sample size, individuals who were not non-Hispanic White, Black, and Hispanic were not included in the analysis. The final analytic sample size was 7900. Detailed survey data and response rates for each U.S. state and survey year are available on the BRFSS website (https://www.cdc.gov/brfss/annual_data/annual_data.htm (accessed on 17th August 2023)).

### 2.2. Study Variables

Discussion of Memory Loss with a Healthcare Provider. If respondents answered in the affirmative to a subjective memory decline question, they were further probed as to whether they or anyone else discussed confusion or memory loss with a healthcare professional. The answer categories included ‘yes’ (1) or ‘no’ (0).

Race/Ethnicity. Self-reported race and ethnicity were categorized in the BRFSS as non-Hispanic White, non-Hispanic Black, non-Hispanic Asian-American, non-Hispanic Pacific Islander, non-Hispanic-American Indian or Alaskan Native, Hispanic or Latino, multiracial, or non-Hispanic other. For the purpose of the analysis, we categorized race into non-Hispanic White (reference), non-Hispanic Black, and Hispanic or Latino.

Primary Healthcare Provider (PCP). We included the availability of a primary healthcare provider with a question that asked, “Do you have one person you think of as your personal doctor or healthcare provider?”; the answer categories were ‘yes’ (1) or ‘no’ (0).

### 2.3. Covariates

The analyses further adjusted for variables that are documented to be associated with subjective memory loss and access to a primary care provider, based on the prior literature [4]. The variables included sex (male and female), age group (50–54; 55–59; 60–64; 65+), highest educational attainment (no college diploma; college diploma), annual household income (<35,000; 35,000 or more), self-reported mental health status (<14 days of poor mental health; 14 or more days of poor mental health), chronic conditions (0; 1; 2+ conditions from a heart attack, angina, stroke, asthma, hypertension, diabetes, chronic kidney disease, COPD, arthritis, cancer), and functional limitation (need of significant assistance with any Activities of Daily Living [ADL] task). For mental health status, respondents were asked to report on average how many days during the past 30 days their mental health was not good, including stress, depression, and problems with emotions. To assess functional limitations, respondents were asked to report whether they needed help for day-to-day activities, including walking, dressing or bathing, and running errands alone. If respondents answered in the affirmative to *any* of these items, we considered them as having an ADL limitation.

### 2.4. Analysis

The analyses used the BRFSS-provided survey weights in order to account for the complex survey design and cellular telephone survey data. The incorporation of survey weights correctly extrapolates the results to national-level estimates. Since residents of the same state share similar environments, we used a clustering option to account for the shared variance within states. In order to test for the main effects of race and PCP, we fit logistic regression models to examine the discussion of memory loss as a function of race, ethnicity, and PCP while controlling for the relevant covariates, survey weights, and state indicators. In order to test for the multiplicative effects, we included an interaction term (in addition to the main effects) between each race category and having a PCP. Each significant interaction term was then tested using the F-test statistics (*p* < 0.05). Although the log-likelihood test is generally conducted to compare the additive model and interaction models, this test was not appropriate for survey estimation results using complex survey weights. To test for the robustness of the final analytic model, several supplementary analyses were conducted. The models included several other covariates, including veteran status, self-reported social support, binge drinking, and leisure time and physical activity. These variables were subsequently omitted from the analyses because they were neither significant nor changed the substantive conclusions. Additional statistical tests, such as the log-likelihood ratio test, revealed that the model with additional covariates did not differ from the final analytic model in terms of the model fit (LR chi-square = 5.86, *p* = 0.06). The analyses also tested alternative specifications of existing covariates, such as six-category income and count variable for chronic conditions. All analyses were conducted using Stata 16.1. For variables with less than 1% missing data, we employed listwise deletion. However, household income had the most amount of missing data because 21% of the sample had missing data for this variable. We imputed the income variable and created 10 imputed datasets.

## 3. Results

The descriptive statistics for core variables are presented in Table 1. The percentages are appropriately weighted using the BRFSS survey weights. Among 7900 individuals who were aged 50 and older with SCD, the majority of respondents were female, non-Hispanic White, and 65 and older. Approximately half of the respondents were unmarried, 45% earned $35,000 or more annually, almost half of the respondents were college-educated, 22% reported 14 or more days of poor mental health, 69% exhibited more than two chronic conditions, and 55% reported needing help with Activities of Daily Living (ADL). Finally, a large majority reported having a primary healthcare provider, and approximately half of the respondents discussed their memory issues with physicians and/or healthcare providers.

Table 2 presents the findings from the multivariate logistic regression analysis. Model 1, where race/ethnicity and healthcare provider were added as independent predictors, shows no significant effects related to race and ethnicity. Model 1 shows that those with a PCP were almost twice as likely to discuss their memory decline with a healthcare provider (OR = 1.98, *p* < 0.001). Models 2 and 3 added interaction terms between each race/ethnicity group and having a PCP. There was a significant interaction effect between Hispanic ethnicity and having a PCP (OR = 0.25, *p* < 0.05). The test of the interaction coefficient was significant (F = 4.65, *p* = 0.03). As graphically depicted in Figure 1, while non-Hispanic respondents who had a PCP had a higher probability of discussing memory issues with a healthcare provider (42% vs. 25%), Hispanic respondents with a PCP were less likely to discuss their memory loss with a healthcare provider (36% vs. 49%). The other multiplicative term (PCP x non-Hispanic Black) was not significant.

## 4. Discussion

Given the high risk of developing dementia in people with cognitive decline and the crucial role of a PCP in facilitating early detection, responsive treatment, and provision of support to people with cognitive decline, it is important to attend to the relationship between SCD and healthcare-seeking behaviors. Of particular interest are racial and ethnic differences in response to SCD and access to healthcare. The current study is the first one to use the sample of individuals with subjective cognitive decline in order to study whether having a usual source of care (i.e., primary care provider) is associated with the discussion of memory loss.

A substantial proportion (94%) of the sample had a usual source of care, which is similar to the national statistic for Medicare beneficiaries (95.6%) [22]. Additionally, a majority of the sample discussed their memory issues with physicians and/or healthcare providers. Given that early signs of cognitive decline or impairment are most often first detected in a primary care setting [10], the findings of the current research also indicate that PCPs of people with SCD play a pivotal role in managing their care as a key point person.

This study also adds to the growing literature on the vital role of the PCP in the provision of services to those with cognitive decline. In the multivariate analyses on the discussion of memory loss with a healthcare provider, discussion of memory decline with a healthcare provider was more likely among those with a PCP. In concordance with findings from other studies [14,17], our study highlights the role of the PCP that precipitates memory-loss-related discussions. Since the PCPs are ideally positioned to observe potential signs of cognitive decline and ask appropriate questions [23], findings also call attention to the development and implementation of effective educational programs or trainings that contribute to improving the clinical competence of the PCP, which may help achieve sustained improvements in dementia diagnosis and disease management. Perhaps regular doctor visits for adults aged 50 and older should include a short questionnaire about cognitive abilities and subjective cognitive decline to explicitly offer an opportunity to discuss these issues.

The highlight of the present analyses was their inclusion of interaction terms between each race/ethnicity group and availability of a healthcare provider in the multivariate models, which demonstrates differing effects of having a PCP as a function of race/ethnicity. Non-Hispanic White respondents with a PCP were more likely to discuss their memory loss issues with a healthcare provider than those without a PCP. However, Hispanics with a PCP were less likely to discuss their memory issues. That is, though early detection and diagnosis of dementia is a public health priority, Hispanic respondents with SCD who had a PCP as usual source of care were less likely to discuss their memory issues than their counterparts. These findings yield useful information in addressing disparities in memory-loss-related communication with a PCP among racial/ethnic groups and further elucidate the racial and ethnic disparity in cognitive health. It could be explained that lack of recognition and health literacy (particularly for dementia) and stigmatizing beliefs about dementia can hinder the healthcare-seeking behaviors for this population [9,24]. According to the result of Adult and Primary Care Physician Surveys conducted by the Alzheimer’s Association [25], level of awareness and understanding of mild cognitive impairment among Hispanic-Americans is the lowest across all racial and ethnic groups surveyed.

In some Hispanic groups and communities, there may be a cultural belief that dementia-related changes are a normal part of the aging process [26,27]. In addition, Hispanic-Americans expressed concern about how others in their community would react to the diagnosis of memory loss, cognitive impairment, and even dementia, and had the perceived stigma of a diagnosis of dementia [27]. Such stigmas surrounding dementia among Hispanic-Americans may make them more hesitant to communicate about their memory loss with a PCP whom they know well, which can result in barriers to prompt cognitive testing, timely diagnosis, and quality of care even though early interventions offer a better opportunity for the management and treatment of dementia.

Along with the expansion of preventive healthcare benefits under the Affordable Care Act (ACA), the Centers for Medicare and Medicaid Services (CMS) have attempted to increase the use of cognitive assessments in routine care by making the Medicare Annual Health Visit (AWV) mandatory [28]. However, as suggested by the findings of the current study, Hispanic-Americans were less likely to discuss memory decline issues in primary care. Responding to the fact that Hispanic-Americans are about 1.5 times more likely than non-Hispanic Whites to have Alzheimer’s or other dementias [25], these findings have practical implications for policymakers and healthcare providers to pay attention to racial/ethnic minorities, particularly Hispanic-Americans, in developing and disseminating culturally appropriate cognitive assessment tools for use, as well as increasing public awareness and health literacy on cognitive health. Additionally, collaborative care models, which include not only physicians and geriatricians, but also social workers, nurses, and medical assistants, can improve care coordination, thereby reducing racial and ethnic health disparity in cognitive health.

Despite the study’s contributions, the interpretations of the findings might be tempered by the study’s limitations. First, BRFSS does not include measures on the quality of care. Studies show that non-White respondents generally receive less appropriate health services (particularly for diabetes and hypertension), express lower levels of healthcare satisfaction, and exhibit less trust towards healthcare providers [29]. Since healthcare provision as a concept is much more than having a primary care provider, future studies should consider both the quantity and quality of care. Additionally, functional limitation was assessed with a single-item question of whether there was need of significant assistance with any Activities of Daily Living task. This limitation is inherent to the secondary analysis of data. Secondly, further studies should investigate the mechanisms linking healthcare utilization, SCD, and cognitive impairment. A meta-analysis of longitudinal studies shows that of those who reported SCD, 14% of individuals develop dementia and 27% of individuals develop MCI [30]. A window of time between SCD and dementia offers an avenue for evidence-based treatments and interventions. Third, while BRFSS offers unique data and a large sample size, longitudinal studies should be conducted in order to examine whether the access and use of healthcare services affect the prevalence and incidence of SCD. There could be potential confounding factors that were not included in the current analyses, including quality of care, the type of care, other types of support, insurance, and other preexisting conditions that might affect the use of primary healthcare services. Fourth, given that the majority of the sample in the BRFSS reported having a PCP, there is an underrepresentation of individuals without access to healthcare or a healthcare provider. Future studies must survey the populations or geographical areas that have limited access to healthcare services in order to further investigate the effects of PCP availability and the discussion of cognitive issues. Lastly, it is important to note that some racial groups with small sample sizes were not represented in the current analyses. Hence, the current findings are not generalizable to underrepresented racial/ethnic groups, such as Native Americans, Pacific Islanders, and multiracial groups. Studies should identify barriers to healthcare among people with SCD in underserved racial ethnic groups.

Despite the limitations, the study offers unique findings about the role of primary care providers for individuals who report SCD while providing nuanced findings on race and ethnicity. SCD can be a clinically meaningful and modifiable indicator of cognitive decline. Though the International SCD Working Group (SCD-I) offers a conceptual framework on SCD, this framework focuses on the detection of preclinical AD rather than SCD. The importance of SCD must be considered when evaluating cognition in a primary care setting, particularly for multiple racial and ethnic groups.

## Figures and Tables

**Figure 1 behavsci-13-00955-f001:**
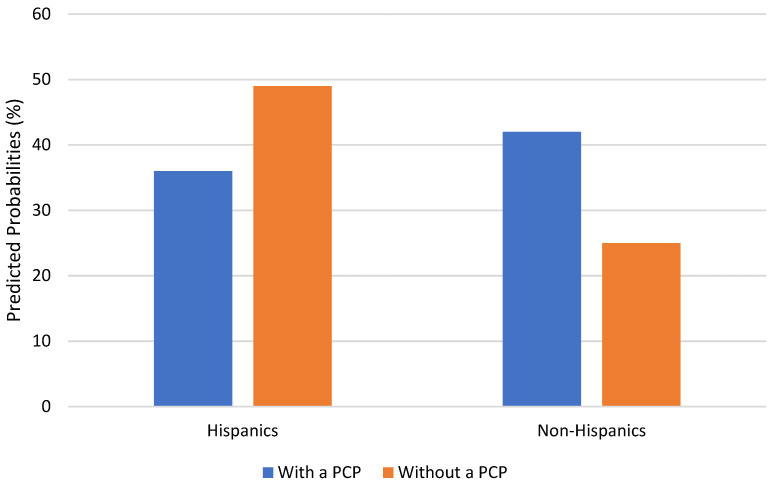
PCP availability and Discussion of Memory issues, by Hispanic Ethnicity.

**Table 1 behavsci-13-00955-t001:** Descriptive Characteristics of the Sample with Subjective Cognitive Decline in the BRFSS (n = 7900).

	%		
	Total with SCD (n = 7900)	Have a Primary Care Provider (PCP)	
		Yes (93%, n = 7347)	No (7%, n = 553)
Age			
50–54	24.9	28.8	22.7
55–59	24.1	26.9	21.7
60–64	21.3	21.1	21.3
65+	29.7	23.2	34.3
Sex			
Male	44.3	41.2	46.9
Female	55.7	58.8	53.1
Race/Ethnicity			
non-Hispanic White	81.7	86.5	82.8
non-Hispanic Black	8.9	9.2	11.1
Hispanic	9.4	4.3	6.1
Education			
Less than college	52.1	42.9	53.9
College graduate	47.9	57.1	46.1
Income			
<35,000	55.3	50.2	57.1
35,000 or more	44.7	49.8	42.9
Marital Status			
Partnered	50.1	49.1	50.0
Unpartnered	49.9	50.9	50.0
Mental Health			
<14 days	78.1	70.6	82.7
14 or more days	21.9	29.4	17.3
Chronic Conditions			
0	2.0	2.0	2.9
1	28.7	20.1	35.9
2+	69.3	77.9	61.2
ADL Limitation			
No help needed	44.6	39.1	49.5
Help needed	55.4	60.9	50.5
Discussed Memory Issues			
Yes	45.8	47.4	28.3
No	54.2	52.6	71.7

**Table 2 behavsci-13-00955-t002:** Logistic Regression Analysis Predicting Discussion of Memory Loss with a Healthcare Provider.

	Baseline Model		Model 1		Model 2		Model 3	
	Odds Ratio	95% CI	Odds Ratio	95% CI	Odds Ratio	95% CI	Odds Ratio	95% CI
Have a PCP	2.27 ***	2.00–2.59	1.98 ***	1.31–2.99	2.19 **	1.40–3.44	2.10 ***	1.66–2.66
Black			0.94	0.70–1.26	0.94	0.71–1.27	0.80	0.36–1.80
Hispanic			0.81	0.48–1.36	3.04	0.97–9.49	0.99	0.76–1.29
PCP * Hispanic					0.25 *	0.70–0.88		
PCP * Black							1.19	0.52–2.72
Age			0.84 ***	0.78–0.90	0.84 ***	0.78–0.90	0.86 ***	0.82–0.89
Female			1.30 **	1.09–1.54	1.29 **	1.09–1.54	1.37 ***	1.24–1.51
College Educated			1.33 **	1.10–1.60	1.33 **	1.11–1.60	1.55 ***	1.40–1.72
Income			1.31 *	1.06–1.62	1.31 *	1.06–1.62	1.05	0.93–1.18
Unmarried			1.16	0.97–1.39	1.16	0.98–1.39	1.06	0.96–1.18
Mental Health			1.64 ***	1.33–2.03	1.64 ***	1.33–2.02	1.79 ***	1.59–2.02
Chronic Conditions			1.60 ***	1.32–1.94	1.60 ***	1.32–1.94	1.44 ***	1.30–1.59
ADL limitation			1.35 **	1.13–1.62	1.35 **	1.13–1.62	1.42 **	1.28–1.57

* *p* < 0.05, ** *p* < 0.01, *** *p* < 0.001.

## Data Availability

Publicly available datasets were analyzed in this study. This data can be found here: https://www.cdc.gov/brfss/annual_data/annual_data.htm.

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
