# Peer review of "Access to Primary Healthcare and Discussion of Memory Loss with a Healthcare Provider in Adults with Subjective Cognitive Decline: Does Race/Ethnicity Matter?"

_behavsci, 2023, doi:10.3390/bs13110955_

Round 1

Reviewer 1 Report

Comments and Suggestions for Authors

Dear Authors,

I had the opportunity to revise your paper, entitled “Access to Primary Healthcare and Discussion of Memory Loss with a Health Care Provider in Adults with Subjective Cognitive Decline: Does Race/ethnicity Matter?”. 

It is basically a cross-sectional study, based on BRFSS (USA), aiming to describe the association between access to primary care and SCD, studying the potential influence of ethnicity.

In the opinion of the Reviewer, the theme is of interest; nevertheless, there are some concerns to be addressed.

Below you can find specific comments:

1)    Introduction: in the second paragraph the Authors cite the link between SCD and dementia without mentioning the role that neuropsychological tests and multidimensional assessment indeed has in this matter, all the more so because in Methods they take into account mental health, chronic conditions, and functional limitations as such. The Reviewer would encourage deepening the section with up-to-date references, such as: doi: 10.1007/s40520-019-01307-2 (about mood), doi: 10.3390/geriatrics8010012 (about cognitive screening), doi: 10.1017/S104161021600106X (about multimorbidity).

2)    Methods – Covariates: please clarify what “mental health” means.

3)    Results – Descriptive statistics: please rephrase the first sentence (line 161).

4)    Results – Descriptive analysis (lines 167-168): please clarify what “needing assistance” means.

5)    Results – Descriptive analysis (lines 168): please clarify what “regular” means.

6)    Table 1: in some cases, the sum is not 100, please check every variable and use commas.

7)    Results: prior giving the adjusted models, the Authors should provide the regression without confounders (that is only with x1=having PCP).

8)    Results: do the Authors follow specific criteria to consider this number of variables as confounders? 

9)    Discussion: further comments would be provided after the address of the previous points.

Regards                                    

Comments on the Quality of English Language

English is fine, just minor editing is required.

Author Response

We sincerely appreciate you for the thoughtful comments on our work. This addresses how we responded to each of the concerns raised. The revised parts in the manuscript are colored in red (Please see the attachment for revised manuscript).  

Reviewer 1

1)    Introduction: in the second paragraph the Authors cite the link between SCD and dementia without mentioning the role that neuropsychological tests and multidimensional assessment indeed has in this matter, all the more so because in Methods they take into account mental health, chronic conditions, and functional limitations as such. The Reviewer would encourage deepening the section with up-to-date references, such as: doi: 10.1007/s40520-019-01307-2 (about mood), doi: 10.3390/geriatrics8010012 (about cognitive screening), doi: 10.1017/S104161021600106X (about multimorbidity).

Response: The following statement, along with the suggested references, has been added.

Previous studies have shown that SCD not only reflects objective cognitive performance and neurological test outcomes [a, b] but also is closely associated with physical and mental health indicators [c, 21].

a. Spano, G.; Caffò, A. O.; Lanciano, T.; Curci, A.;& Bosco, A.  Visuospatial/executive abilities and mood affect the reliability of a subjective memory complaints measure. Aging Clin Exp Res 2020, 32, 1317-1326.

b. Salis, F.; Costaggiu, D.; & Mandas, A.Mini-mental state examination: optimal cut-off levels for mild and severecognitive impairment. Geriatrics 2023, 8, 12.

c. Pedro, M. C.; Mercedes, M. P.; Ramón, L. H.; &Borja, M. R. Subjective memory complaints in elderly: relationship with healthstatus, multimorbidity, medications, and use of services in a population-basedstudy. Int psychogeriatrics, 2016, 28, 1903-1916.

21. Jang, Y.; Haley, W. E.; Choi, E. Y.; & Franco, Y. Racial/ethnicdifferences in correspondence between subjective cognitive ratings andcognitive impairment. The Am J Geriatr Psychiatry 2022, 30,627-635.

2)    Methods – Covariates: please clarify what “mental health” means.

Response: Mental health is self-reported; respondents were asked to report average days during the past 30 days that their mental health was not good, including stress, depression, and problems with emotions. The categories were ‘none to less than 14 days’ or ’14 or more days.’ We updated the language to clarify that it’s self-reported poor mental health status.

self-reported mental health status (<14 days of poor mental health; 14 or more days of poor mental health).

For mental health status, respondents were asked to report average days during the past 30 days that their mental health was not good, including stress, depression, and problems with emotions

3)    Results – Descriptive statistics: please rephrase the first sentence (line 161).

Response: We rephrased the first sentence to start with the descriptive statistics.

The descriptive statistics for core variables are presented in Table 1. The percentages are appropriately weighted by the BRFSS survey weights. Among 7,900 individuals who were aged 50 and older with SCD, majority respondents were female, non-Hispanic White, and 65 and older.

4)    Results – Descriptive analysis (lines 167-168): please clarify what “needing assistance” means.

Response: We changed ‘needing assistance’ to ‘needing help with ADL tasks.’

5)    Results – Descriptive analysis (lines 168): please clarify what “regular” means.

Response: We changed ‘regular’ to ‘primary,’ as in ‘primary care provider.’

6)    Table 1: in some cases, the sum is not 100, please check every variable and use commas.

Response: We found that percentages for the age groups did not add up to 100 percent due to rounding. We corrected the numbers accordingly.

7)    Results: prior giving the adjusted models, the Authors should provide the regression without confounders (that is only with x1=having PCP).

Response: In response to this comment, we added columns for the baseline model.

8)    Results: do the Authors follow specific criteria to consider this number of variables as confounders? 

Response: The covariates were selected because 1) prior studies on subjective cognitive decline have used the covariates and 2) the covariates in the main analyses have been chosen after conducting several supplementary analyses using other covariates. We briefly discussed this in the analysis section.

The analyses further adjusted for variables that are documented to be associated with subjective memory loss and access to primary care provider, based on the prior literature [4].

To test for the robustness of the final analytic model, several supplementary analyses were conducted. The models included several other covariates, including veteran status and self-reported social support. These variables were subsequently omitted from the analyses because they were neither significant nor changed the substantive conclusions. The analyses also tested alternative specifications of existing covariates, such as six-category income and count variable for chronic conditions.

9)    Discussion: further comments would be provided after the address of the previous points.

Response: In light of this comment, we discussed other potential confounding factors in the discussion section.

There could be potential confounding that was not included in the current analyses including quality of care, the type of care, other types of support, insurance, and other preexisting conditions that might affect the use of primary healthcare services.

Again, we sincerely appreciate your thoughtful comments. We believe that our manuscript is substantially improved by these revisions. We hope that you will find it suitable for publication in the Special Issue "Cognitive, Emotional, and Psychological Changes in Healthy and Pathological Aging" of Behavioral Sciences

Sincerely yours,

Reviewer 2 Report

Comments and Suggestions for Authors

The work used the ‘Behavioral Risk Factor Surveillance System’ database (2019-2020) to address a critical issue such as the ethnic disparity in memory loss-related communication with healthcare providers in a primary care setting, with the Hispanic American population in the US (where the study is done) as those that, despite having a usual source of healthcare (found to be a positive factor), were likely to do so. This is discussed according to known differences in self-awareness and cultural beliefs, among other potential reasons. This particular finding is relevant because this population is about 1.5 times more likely to have Alzheimer’s or other dementias than non-Hispanic whites. The authors also highlight the link between SCD, quality of life, global health, and Alzheimer’s disease, and the key role of primary care settings in the provision of timely diagnosis. Limitations of the study are well identified.

One of the most interesting things / contributions about the present work is the fact that ‘behavior’ (and its interaction with ethnicity/culture) is usually underestimated in the field of dementia, and most literature focuses on biomedical and neuropsychological tools.

My major point of concern is that the title hides the main finding, ‘the unique findings’ (as referred by the authors) about the role of PCP and the lower communication of HA to PCP. Thus, at a glance, in a search, it would seem more like a ‘conceptual review’ than research that yields a ‘unique finding’, so it will not contribute to raising awareness on the “importance of SCD needs to be considered when evaluating cognition in a primary care setting, putting special attention to underreporting/communication of Hispanic Americans, a behavior/scenario that may enhance their already higher risk to develop AD or other dementias. Please, reconsider; this focus/unique finding is lost in the current title. As a parallel scenario, see, for instance, DOI: 10.2174/1567205018666211116093747

The abstract could also be improved in this respect. Please, note that half of the abstract is just to refer to the scenario (lines 11 to 18 out of 11 to 25), and only two sentences to highlight the ‘unique findings’

The sample population, 7,900 individuals from 47 (out of 52 states) where the SCD module is implemented, gives the data a predictive value or national estimates, as stated by the authors, thanks to the incorporation of survey weights.

Since the sociologist team 2 of 3 members' affiliations are in Texas and Southern California, it is guessed that the present research work is wisely born to provide scientific data to target and solve a current observed problem and need.

Some other comments, done in a positive manner that would improve the quality of the present work are:

Lines 114-116. Data set. Minorities have been excluded from the analysis due to statistical power reasons. However, the authors should include that as a limitation and provide their opinion on their expectations in these other populations.

Lines 129-131. Just a comment: It is interesting that the survey also recorded the opinion of PCP. Maybe some discussion about it would be appropiate, a part of the key role (maybe refering to the need to increase such awareness among clinicians).

Line 137. Please, indicate how mental health is defined, as for the ‘time’ description, it seems that it refers to ‘mental well-being’ more than to ICD-11 or DMS-V mental health disorders classification.
Similarly, in Lines 140-141. Please, indicate which scale for DLA was used or if any, how was it ranked (absence/presence, etc). If both cases were those already established by the database, please discuss these limitations and the authors expectations.

Table 1. Please add the variables (n) or (%) as corresponds after each descriptive, and do a yes/no column distribution so it is easier to read. That is: Total with SCD (n), but percentages in the other two columns. As a suggestion, for a better presentation of data, add a single heading for the variable “Discussed memory problem with a Health Care Provider” and then two columns, one for YES (% of 7,347), NO (% of n=553). This will also allow for a better size of the table, not so spread with blank areas between results.

Figure 1. Please, present the figure plot in a way that is not obvious that it is a direct capture of excel. Just removing the frame would help.

Author Response

We sincerely appreciate you for the thoughtful comments on our work. This addresses how we responded to each of the concerns raised. The revised parts in the manuscript are colored in red (Please see the attachment for revised manuscript).  

Reviewer 2

  • The work used the ‘Behavioral Risk Factor Surveillance System’ database (2019-2020) to address a critical issue such as the ethnic disparity in memory loss-related communication with healthcare providers in a primary care setting, with the Hispanic American population in the US (where the study is done) as those that, despite having a usual source of healthcare (found to be a positive factor), were likely to do so. This is discussed according to known differences in self-awareness and cultural beliefs, among other potential reasons. This particular finding is relevant because this population is about 1.5 times more likely to have Alzheimer’s or other dementias than non-Hispanic whites. The authors also highlight the link between SCD, quality of life, global health, and Alzheimer’s disease, and the key role of primary care settings in the provision of timely diagnosis. Limitations of the study are well identified. One of the most interesting things / contributions about the present work is the fact that ‘behavior’ (and its interaction with ethnicity/culture) is usually underestimated in the field of dementia, and most literature focuses on biomedical and neuropsychological tools.

Response: We appreciate the reviewer’s positive remarks.

  • My major point of concern is that the title hides the main finding, ‘the unique findings’ (as referred by the authors) about the role of PCP and the lower communication of HA to PCP. Thus, at a glance, in a search, it would seem more like a ‘conceptual review’ than research that yields a ‘unique finding’, so it will not contribute to raising awareness on the “importance of SCD needs to be considered when evaluating cognition in a primary care setting, putting special attention to underreporting/communication of Hispanic Americans, a behavior/scenario that may enhance their already higher risk to develop AD or other dementias. Please, reconsider; this focus/unique finding is lost in the current title. As a parallel scenario, see, for instance, DOI: 10.2174/1567205018666211116093747

Response: We agree with the Reviewer that our study title does not fully capture the unique finding on Hispanics and have considered a new title. After careful discussions, however, we have decided to keep the original title. The primary reason is that our approach with race/ethnicity was explorative rather than theory- or hypothesis-driven. We would like to keep the explorative nature of the study and present the finding on Hispanics as a product of our investigation. To support and validate our major finding, we have elaborated the dementia-related cultural beliefs, health service use, and health communications among Hispanics in the discussion section.  

  • The abstract could also be improved in this respect. Please, note that half of the abstract is just to refer to the scenario (lines 11 to 18 out of 11 to 25), and only two sentences to highlight the ‘unique findings’

Response: We thank the reviewer for this comment. We revised the abstract in order to highlight the unique findings on race/ethnic differences in having a PCP and healthcare utilization among individuals with SCD.

Abstract: Since subjective cognitive decline (SCD) manifests before the clinical and irreversible onset of dementia, efforts to address SCD are imperative for the early identification and prevention of dementia. Among people who actually experience SCD, having a usual source of care may be an important catalyst for addressing the memory issues. The present study investigates the relationship between having a usual source of care and the discussion of SCD with healthcare provider, while taking into account racial and ethnic differences. The pooled 2019-2020 Behavioral Risk Factor Surveillance System (BRFSS) from 47 states in the U.S. was used. The sample included 7,900 individuals aged 50 and older who reported significant cognitive decline. Using logistic regression with state-level clustering, we found that those with a usual source of healthcare was almost twice as more likely to discuss memory loss with a healthcare provider. In order to test for the multiplicative effects, we included an interaction term between each race category and having a primary care provider (PCP). There was a significant interaction effect between Hispanic ethnicity and having a PCP (OR=0.25, p<.05). While non-Hispanic respondents having a PCP had a higher probability of discussing memory issues with a healthcare provider (42% vs. 25%), Hispanic respondents with a PCP were less likely to discuss their memory loss with a healthcare provider (36% vs. 49%). The study adds to the growing literature on the vital role of the primary care service in provision of service to those with cognitive decline, while revealing ethnic disparity in memory loss-related communication with healthcare providers in a primary care setting.

  • The sample population, 7,900 individuals from 47 (out of 52 states) where the SCD module is implemented, gives the data a predictive value or national estimates, as stated by the authors, thanks to the incorporation of survey weights.

Response: Correct- 5 states were removed from the analyses due to the small sample size.

  • Since the sociologist team 2 of 3 members' affiliations are in Texas and Southern California, it is guessed that the present research work is wisely born to provide scientific data to target and solve a current observed problem and need.

Response: We appreciate the reviewer’s positive comment.

  • Some other comments, done in a positive manner that would improve the quality of the present work are: Lines 114-116. Data set. Minorities have been excluded from the analysis due to statistical power reasons. However, the authors should include that as a limitation and provide their opinion on their expectations in these other populations.

Response: In light of this comment, we noted the lack of ‘other’ racial groups as a limitation of the study.

Lastly, it is important to note that some racial groups with small sample sizes were not represented in the current analyses. Hence, the current findings are not generalizable to underrepresented racial/ethnic groups such as Native Americans, Pacific Islanders, and multiracial groups. Studies should identify barriers to healthcare among people with SCD of underserved racial ethnic groups.

  • Lines 129-131. Just a comment: It is interesting that the survey also recorded the opinion of PCP. Maybe some discussion about it would be appropiate, a part of the key role (maybe refering to the need to increase such awareness among clinicians).

Response: We concur that having a primary care provider does not indicate the quality of such care. Future research projects must address the quality of care (such as the relationship with PCP) and health outcomes among individuals with SCD.

First, BRFSS does not include measures on the quality of care. Studies show that non-White respondents generally receive less appropriate health services (particularly for diabetes and hypertension), express lower levels of healthcare satisfaction, and exhibit less trust towards healthcare providers [26]. Since healthcare provision as a concept is much more than having a primary care provider, future studies should consider both the quantity and quality of care.

  • Line 137. Please, indicate how mental health is defined, as for the ‘time’ description, it seems that it refers to ‘mental well-being’ more than to ICD-11 or DMS-V mental health disorders classification.

Response: Mental health is self-reported; respondents were asked to report average days during the past 30 days that their mental health was not good, including stress, depression, and problems with emotions. The categories were ‘none to less than 14 days’ or ’14 or more days.’ We updated the language to clarify that it’s self-reported poor mental health. This is a crude measure, particularly compared to other measurement such as Center for Epidemiological Studies – Depression (CES-D).

self-reported mental health status (<14 days of poor mental health; 14 or more days of poor mental health).

For mental health status, respondents were asked to report average days during the past 30 days that their mental health was not good, including stress, depression, and problems with emotions.

  • Similarly, in Lines 140-141. Please, indicate which scale for DLA was used or if any, how was it ranked (absence/presence, etc). If both cases were those already established by the database, please discuss these limitations and the authors expectations.

Response: This was also addressed in the limitation of the discussion section.

To assess functional limitation, respondents were asked to answer the following question, “during the past 12 months, confusion or memory loss, you given up/need assistance with day-to-day household activities or chores you used to do, such as cooking, cleaning, taking medications, driving, or paying bills?”

In addition, functional limitation was assessed with a single-item question whether there is need of significant assistance with any Activities of Daily Living task or not. This limitation is inherent to the secondary analysis of data.

  • Table 1. Please add the variables (n) or (%) as corresponds after each descriptive, and do a yes/no column distribution so it is easier to read. That is: Total with SCD (n), but percentages in the other two columns. As a suggestion, for a better presentation of data, add a single heading for the variable “Discussed memory problem with a Health Care Provider” and then two columns, one for YES (% of 7,347), NO (% of n=553). This will also allow for a better size of the table, not so spread with blank areas between results.

Response: The table was revised per reviewer’s suggestion.

  • Figure 1. Please, present the figure plot in a way that is not obvious that it is a direct capture of excel. Just removing the frame would help.

Response: The graph is integrated into Words, but we removed the border, per reviewer’s suggestion.

Again, we sincerely appreciate your thoughtful comments. We believe that our manuscript is substantially improved by these revisions. We hope that you will find it suitable for publication in the Special Issue "Cognitive, Emotional, and Psychological Changes in Healthy and Pathological Aging" of Behavioral Sciences

Sincerely yours,

Reviewer 3 Report

Comments and Suggestions for Authors

I think it is a well-conducted study that looked at the relationship between having a usual source of care and discussing SCD with a health care provider, while taking into account racial and ethnic differences. Therefore, I recommend its publication after these corrections:

Methods

Analysis: Please, point to the strength of the evidence p value.

Results

Descriptive statistics: Give a title to the results, eliminate "descriptive statistics". Line 160

You need to reduce the description to percentages, which you can find in the table. Line 161-172

The description of table 2 is unclear, rephrase the discussion. Line 177-188

Discussion 

Remove the percentages in the discussion and rephrase the discussion.

Comments on the Quality of English Language

The use of English should be improved to make sentences clearer.

Author Response

We sincerely appreciate you for the thoughtful comments on our work. This addresses how we responded to each of the concerns raised. The revised parts in the manuscript are colored in red (Please see the attachment for revised manuscript).  

  • I think it is a well-conducted study that looked at the relationship between having a usual source of care and discussing SCD with a health care provider, while taking into account racial and ethnic differences. Therefore, I recommend its publication after these corrections:

Methods. Analysis: Please, point to the strength of the evidence p-value.

Response: We added a reference p-value for the statistical significance.

  • Results

Descriptive statistics: Give a title to the results, eliminate "descriptive statistics". Line 160

Response: In response to this comment, we eliminated the sub-heading “descriptive statistics.”

  • You need to reduce the description to percentages, which you can find in the table. Line 161-172

Response: We attempted to explain the descriptive statistics in text. In light of this comment, we shortened the description of percentages.

The descriptive statistics for core variables are presented in Table 1. The percentages are appropriately weighted by the BRFSS survey weights. Among 7,900 individuals who were aged 50 and older with SCD, majority respondents were female, non-Hispanic White, and 65 and older.

  • The description of table 2 is unclear, rephrase the discussion. Line 177-188

Response: We clarified the discussion of Table 2 in order to highlight the main findings about Hispanic ethnicity-specific findings.

Table 2 presents the findings from multivariate logistic regression analysis. Model 1, where race/ethnicity and healthcare provider were added as independent predictors, shows no main effects of race and ethnicity. Model 1 shows that those with a PCP were almost twice as likely to discuss their memory decline with a healthcare provider (OR=1.98, p<.001). Model 2 and 3 added interaction terms between each race/ethnicity group and having a PCP. There was a significant interaction effect between Hispanic ethnicity and having a PCP (OR=0.25, p<.05). The test of the interaction coefficient was significant (F=4.65, p=0.03). As graphically depicted in Figure 1, while non-Hispanic respondents having a PCP had a higher probability of discussing memory issues with a healthcare provider (42% vs. 25%), Hispanic respondents with a PCP were less likely to discuss their memory loss with a healthcare provider (36% vs. 49%). The other multiplicative term (PCP x non-Hispanic Black) was not significant.

  • Discussion 

Remove the percentages in the discussion and rephrase the discussion.

Response: We removed the percentages in the discussion, except the sentence that compares the current findings to the national statistics. This comparison highlights that the percentages from the current study are comparable to other nationally representative data.

Again, we sincerely appreciate your thoughtful comments. We believe that our manuscript is substantially improved by these revisions. We hope that you will find it suitable for publication in the Special Issue "Cognitive, Emotional, and Psychological Changes in Healthy and Pathological Aging" of Behavioral Sciences

Sincerely yours,

Round 2

Reviewer 1 Report

Comments and Suggestions for Authors

Dear Authors,

the Reviewer thanks you for the reply and believes the Manuscript is improved.

Anyway, the Reviewer also believes that there are still crucial points to be addressed.

Below you can find specific comments:

1)    Analysis: the robustness of the model should not be tested as the Authors did, since this method, although extremely common in the literature, is liable to inherent imprecision. The Author should retest the model according to the first criterion they followed (previous literature), excluding since from the beginning clearly not associated variables, and eventually test the robustness with appropriate methods (did the Authors think on ROC curves, for example?)

2)    Results (page 5, lines 190-191): this point has not been addressed. The Authors should clarify what “needing help with ADL” means: in other words, the Authors should provide the threshold they used to consider a “needing help person”. It can be placed also in Covariates, if the Authors prefer.

3)    Discussion: further comments would be provided after the address of the previous points.

4)    References: the Authors should incorporate the added citations.

Regards

Author Response

Dear Reviewer 1, 

We sincerely appreciate you for the thoughtful comments on our work. This addresses how we responded to each of the concerns raised. The revised parts in the manuscript are colored in red as well.

1)    Analysis: the robustness of the model should not be tested as the Authors did, since this method, although extremely common in the literature, is liable to inherent imprecision. The Author should retest the model according to the first criterion they followed (previous literature), excluding since from the beginning clearly not associated variables, and eventually test the robustness with appropriate methods (did the Authors think on ROC curves, for example?)

Response: Thank you for the insightful comments. We concur that the common practice in the literature, just like any other robustness testing method, is liable to imprecision (e.g., false positive). Though not detailed in the text, we compared various indices for goodness of fit such as pseudo R-squared, log likelihood, and chi-square. Not only did the additional covariates remain non-significant, they also did not improve the fit in any meaningful ways. Finally, including additional covariates did not change the conclusions about the main hypothesis. Although ROC analyses are more common with clinical outcomes, we additionally conducted the ROC analyses. The table below compares the fit indices.

Final analytic model

Supplementary model

Pseudo R2

0.0712

0.0728

Chi-square

10213.42 (p=0.12)

10543.48 (p=0.10)

Area under ROC

0.6748

0.6756

Additionally, we conducted the log likelihood ratio (lrtest) to compare the nested models and the models were not significantly different (LR chi=5.86, p=0.06). All of these indicate that the final analytic model is not different from the supplementary model with additional covariates. The additional covariates included veteran status, perceived social support from friends and family, binge drinking, and leisure time physical activities. We believe that other covariates, such as chronic conditions, general mental health, and functional limitation are accounting for the potential covariates not included in the analyses.

The models included several other covariates, including veteran status, self-reported social support, binge drinking, and leisure time physical activity.

Additional statistical tests such as Log likelihood ratio test revealed that the model with additional covariates did not differ from the final analytic model in terms of the model fit (LR chi-square=5.86, p=0.06).

2)    Results (page 5, lines 190-191): this point has not been addressed. The Authors should clarify what “needing help with ADL” means: in other words, the Authors should provide the threshold they used to consider a “needing help person”. It can be placed also in Covariates, if the Authors prefer.

Response: We appreciate this comment. Just to clarify, the question was self-reported, and the respondents reported whether they needed help or not for day-to-day activities, including walking, dressing or bathing, and running errands alone. If respondents answered affirmatively to any of these items, we considered them having an ADL limitation. We clarified it in the main body.

To assess functional limitation, respondents were asked to reported whether they needed help or not for day-to-day activities, including walking, dressing or bathing, and running errands alone. If respondents answered affirmatively to any of these items, we considered them having an ADL limitation.

3)    Discussion: further comments would be provided after the address of the previous points.

Response: This was added in the limitation of the discussion section

In addition, functional limitation was assessed with a single-item question whether there is need of significant assistance with any Activities of Daily Living task or not. This limitation is inherent to the secondary analysis of data.

4)    References: the Authors should incorporate the added citations

Response: We incorporated the added citations in order.

Sincerely yours,

Round 3

Reviewer 1 Report

Comments and Suggestions for Authors

Dear Authors,

the Reviewer thanks you for the reply and believes the Manuscript is greatly improved, especially in its methodology.

Regards